

# Honeycomb rare-earth magnets
# with anisotropic exchange interactions

**Zhu-Xi Luo[1,2] and Gang Chen[3,4⋆]**

**1** Kavli Institute for Theoretical Physics, University of California, Santa Barbara, CA 93106
**2** Department of Physics and Astronomy, University of Utah, Salt Lake City, UT 84102
**3** Department of Physics and HKU-UCAS Joint Institute for Theoretical and Computational
Physics at Hong Kong, The University of Hong Kong, Hong Kong, China
**4** State Key Laboratory of Surface Physics and Department of Physics,
Fudan University, Shanghai 200433, China

⋆ gangchen.physics@gmail.com

## Abstract

We study the rare-earth magnets on a honeycomb lattice, and are particularly interested in the experimental consequences of the highly anisotropic spin interaction due to the spin-orbit entanglement. We perform a high-temperature series expansion using a generic nearest-neighbor Hamiltonian with anisotropic interactions, and obtain the heat capacity, the parallel and perpendicular spin susceptibilities, and the magnetic torque coefficients. We further examine the electron spin resonance linewidth as an important signature of the anisotropic spin interactions. Due to the small interaction energy scale of the rare-earth moments, it is experimentally feasible to realize the strong-field regime. Therefore, we perform the spin-wave analysis and study the possibility of topological magnons when a strong field is applied to the system. The application and relevance to the rare-earth Kitaev materials are discussed.



# 1 Introduction

Spin liquid candidates are often being searched among geometrically frustrated systems, such as triangular [1], kagomé [2] or pyrochlore [3] lattices. This is quite reasonable as the geometrical frustration could lead to a large number of degenerate or nearly-degenerate classical ground states for commonly studied Heisenberg models and thus enhance quantum fluctuations when the quantum effects are included. However, the destabilization of simple magnetically ordered states and driving a disordered one can happen even on unfrustrated lattices, by exploiting the power of anisotropic interactions [4]; the Kitaev honeycomb model [5] is a representative example of the latter. Besides being an academic interest, anisotropic spin interactions are also inevitable in realistic magnetic materials, especially those with heavy atoms. A large number of spin liquid candidates are known experimentally to possess a significant spin-orbit coupling, leading to rather anisotropic spin interactions [6–11]. Beyond the current interest in the spin liquid physics, understanding the relationship between the magnetic properties and the anisotropic spin interactions is a frontier topic in the field of quantum magnetism.

The most commonly studied anisotropic magnets on unfrustrated lattices are the $4d/5d$ magnets [11, 12] that include the honeycomb iridates and RuCl$_3$. Because of the possible relevance to Kitaev physics, these materials were referred as Kitaev materials. Due to the spatial extension of the $4d/5d$ electron wavefunctions, the exchange interactions between the local moments are usually beyond the nearest neighbors. Moreover, the iridates often suffer from a strong neutron absorption such that the data-rich neutron scattering measurement can be difficult. In comparison, the rare-earth family has the advantages of much stronger spin-orbit couplings and much more localized $4f$ orbitals [13–15], and the exchange interactions often restrict to first neighbors. This makes the understanding of the modeling Hamiltonian more accessible. In addition, the rare-earth magnets do not have the neutron absorption issue that prevails in iridates. Furthermore, their smaller energy scales allow for the possibility to quantitatively understand their Hamiltonian through the external magnetic fields. However, rare-earth materials have only been well-investigated on frustrated lattices [9, 16].

In this paper, we will study the rare-earth magnets on the unfrustrated honeycomb lattice, and pursue an understanding of the experimental consequence of the spin-orbital entanglement on the honeycomb structure. We start by exploring the thermodynamic properties of a generic model with the nearest neighbor interactions. It is well-known that the anisotropic exchange couplings could appear in the temperature dependence of the thermodynamic quantities such as the specific heat, spin susceptibility [17] and magnetotropic coefficients [18,19]. Especially for the spin susceptibility and magnetic torque, magnetic fields along different directions induce magnetization of different magnitudes, leading to the anisotropic spin susceptibility [20–25] and the angular dependence of the magnetic torque [26–29], and providing a natural detection of the intrinsic spin anisotropy in the system. To go beyond the thermody-

namic properties, we further consider the electron spin resonance (ESR) measurement [30,31] of the system. The ESR measurement turns out to be a very sensitive probe of the magnetic anisotropy and is especially useful for the study of the strong spin-orbit-coupled quantum materials, and we compute the ESR linewidth to reveal the intrinsic spin anisotropy of the spin interactions.

Due to the small energy scale of the interaction between the rare-earth local moment, it is ready to apply a small magnetic field in the laboratory to change the magnetic state into a fully polarized one. For such a simple product state, the magnetic excitation can be readily worked out from the linear spin wave theory. We further consider the spin wave spectrum and explore the possibility of topological magnons [32–39]. We find the magnon spectrum supports non-trivial topological band structure. This feature can be manifested in thermal Hall transport measurements.

The remaining parts of the paper are organized as follows. In Sec. 2, we introduce the nearest-neighbor spin Hamiltonian, followed by the high-temperature analysis of heat capacity, spin susceptibilities and magnetic torque coefficient in Sec 3. Then we consider the ESR and calculate the influence of anisotropy on the ESR linewidth in Sec. 4. Next the linear spin wave theory of the system is exploited under strong external fields in Sec. 5 and the aspect of topological magnons is discussed. Finally in Sec. 6 we comment on possible materials YbCl$_3$ and TlYbS$_2$.

## 2   Model

We begin with the following microscopic spin model, that is the most general nearest neighbor Hamiltonian on a honeycomb lattice with the (usual) *Kramers* doublet effective spin-1/2 local moments [6,10,15,40],

$$
\begin{aligned}
H \;=\; & \sum_{\langle ij \rangle} J_{zz} S_i^z S_j^z + J_{\pm}(S_i^+ S_j^- + S_i^- S_j^+) + J_{\pm\pm}(\gamma_{ij} S_i^+ S_j^+ + \gamma_{ij}^* S_i^- S_j^-) \\
& + \; J_{\pm z}[(\gamma_{ij}^* S_i^+ S_j^z + \gamma_{ij} S_i^- S_j^z) + \langle i \leftrightarrow j \rangle],
\end{aligned}
\tag{1}
$$

with $\gamma_{ij}$ taking $e^{2i\pi/3}$, $e^{-2i\pi/3}$, and 1 on the bonds along $\mathbf{a}_1$, $\mathbf{a}_2$, $\mathbf{a}_3$ directions respectively, as shown in Fig. 1. The first two terms give the usual XXZ Hamiltonian, while the latter two terms are bond-dependent and constitute the spin-orbit interaction. The spin components are defined in the global coordinate system in Fig. 1. This is possible because the system is planar and has an unique rotational axis. This differs from the rare-earth pyrochlore materials where the spins are often defined in the local coordinate system for each sublattice. This model applies to the rare-earth local moment such as the Yb$^{3+}$ ion. For non-Kramers doublet like Pr$^{3+}$ or Tb$^{3+}$ ion, the $J_{\pm z}$ term is not allowed by symmetry, and the model becomes further simplified. In fact, a non-Kramers doublet based rare-earth honeycomb magnet arises from the triangular lattice magnet TbInO$_3$ after 1/3 of the Tb$^{3+}$ ions becomes inactive magnetically [41]. For the rare-earth local moments, the $4f$ electrons are much localized, and most often, one only needs to consider the nearest-neighbor interactions, and occasionally, one would like to include the further neighbor dipole-dipole interactions. In contrast, for the $4d/5d$ systems, one may need to worry about further neighbor exchange interactions because of the large spatial extension of the electron wavefunctions.

An alternative and often used parametrization of the Hamiltonian is that of the $J$-$K$-$\Gamma$-$\Gamma'$

model [42]:

$$
\begin{aligned}
H =\ & \sum_{\langle ij\rangle \in \alpha\beta(\gamma)} \left[ J\mathbf{S}_i \cdot \mathbf{S}_j + K S_i^\gamma S_j^\gamma + \Gamma(S_i^\alpha S_j^\beta + S_i^\beta S_j^\alpha) \right] \\
& + \Gamma' \sum_{\langle ij\rangle \in \alpha\beta(\gamma)} \left( S_i^\alpha S_j^\gamma + S_i^\gamma S_j^\alpha + S_i^\beta S_j^\gamma + S_i^\gamma S_j^\beta \right),
\end{aligned}
\tag{2}
$$

where $\alpha, \beta, \gamma$ take values in $\{x', y', z'\}$. In the latter coordinate system, our unit vectors of Fig. 1 can be expressed by $\hat{x} = (-1, -1, 2)/\sqrt{6}$, $\hat{y} = (1, -1, 0)/\sqrt{2}$ and $\hat{z} = (1, 1, 1)/\sqrt{3}$. The spin components in the above equation are

$$
\begin{aligned}
S^{x'} &= -\frac{\sqrt{6}}{6} S_x + \frac{\sqrt{2}}{2} S_y + \frac{\sqrt{3}}{3} S_z, \\
S^{y'} &= -\frac{\sqrt{6}}{6} S_x - \frac{\sqrt{2}}{2} S_y + \frac{\sqrt{3}}{3} S_z, \\
S^{z'} &= \frac{\sqrt{6}}{3} S_x + \frac{\sqrt{3}}{3} S_z.
\end{aligned}
\tag{3}
$$

Furthermore, we have used the notation that $\alpha\beta(\gamma)$ specifies a bond parallel (or anti-parallel) to the vector $\hat{\alpha} - \hat{\beta}$, or simply a bond of type $\gamma$. Here $J$, $K$ and $\Gamma$ are the Heisenberg, Kitaev and symmetric off-diagonal exchange, respectively, and $\Gamma'$ parametrizes the trigonal distortion. The coupling constants in Eq. (1) and (2) are related by the following equation

$$
J = \frac{4}{3}J_\pm - \frac{2\sqrt{2}}{3}J_{\pm z} - \frac{2}{3}J_{\pm\pm} + \frac{1}{3}J_{zz}, \qquad K = 2\sqrt{2}J_{\pm z} + 2J_{\pm\pm},
\tag{4}
$$

$$
\Gamma = -\frac{2}{3}J_\pm - \frac{2\sqrt{2}}{3}J_{\pm z} + \frac{4}{3}J_{\pm\pm} + \frac{1}{3}J_{zz}, \qquad \Gamma' = -\frac{2}{3}J_\pm + \frac{\sqrt{2}}{3}J_{\pm z} - \frac{2}{3}J_{\pm\pm} + \frac{1}{3}J_{zz}.
\tag{5}
$$

The Hamiltonian in Eq. (1) can also be used to describe the general exchange interaction between the higher spin local moments for the honeycomb magnets after some modification. The differences are explained in details in the Appendix A.

## 3 Thermodynamics

The highly anisotropic nature of the exchange interaction first impacts the thermodynamic properties of the system. Here we explicitly calculate the specific heat and the magnetic susceptibilities of the system from the generic exchange Hamiltonian, with details presented in Appendix B. Using the high-temperature series expansion [43, 44], we find the heat capacity

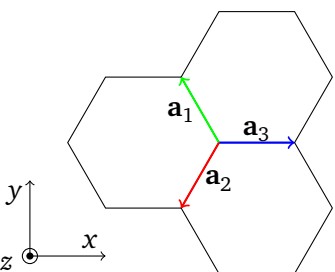

Figure 1: The honeycomb lattice with three different types of bonds and our choice of the global coordinate system.

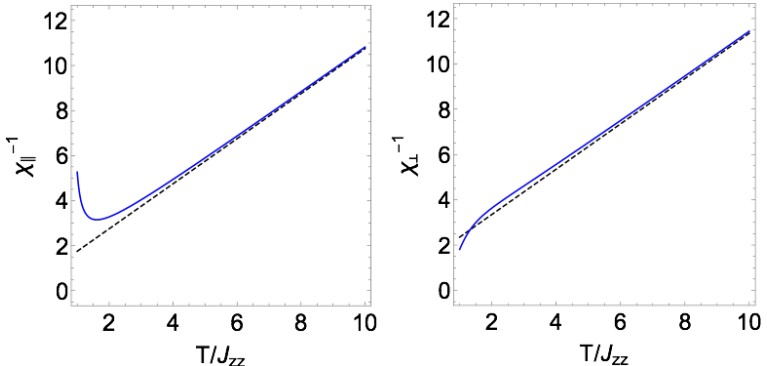

Figure 2: Susceptibilities versus temperature. The dashed black lines show the Curie-Weiss law, and the solid blue lines plot the inverse of equation (9). The parameters are chosen to be $J_{zz} = 1$, $J_\pm = 0.9$, $J_{\pm\pm} = 0.2$, $J_{\pm z} = 0.1$. The susceptibilities in the plot, $\chi_\parallel$ and $\chi_\perp$, are in units of $\mu_0\mu_B^2 g_\parallel^2/4k_B$ and $\mu_0\mu_B^2 g_\perp^2/4k_B$, respectively.

to be

$$C = \frac{3J_0^2}{2k_B T^2} - \frac{27J_0^4}{8k_B^3 T^4}, \tag{6}$$

where $k_B$ is the Boltzmann constant, and

$$J_0^2 \equiv \frac{1}{16}J_{zz}^2 + \frac{1}{2}(J_\pm^2 + J_{\pm\pm}^2 + J_{\pm z}^2). \tag{7}$$

Due to the spin-orbit entanglement, the coupling of the local moment to the external magnetic field is also anisotropic. The Landé factors $g$'s are different for the in-plane and out-plane magnetic fields, and the Zeeman coupling is given as

$$H_Z = -\mu_0\mu_B \sum_i \left[ g_\perp(h_x S_i^x + h_y S_i^y) + g_\parallel h_\parallel S_i^z \right], \tag{8}$$

$h_x, h_y$ are the in-plane components of the external magnetic field, and $h_\parallel$ the z-component. $\mu_0$ is the vacuum permeability and $\mu_B$ the Bohr magneton. Again using high-temperature series expansion, we compute the parallel and perpendicular spin susceptibilities up to $\mathcal{O}(T^{-3})$

$$\begin{aligned}
\chi_\parallel &= \frac{\mu_0\mu_B^2 g_\parallel^2}{4k_B T}\left(1 - \frac{3J_{zz}}{4k_B T} - \frac{J_\pm^2}{2k_B^2 T^2} - \frac{J_{\pm\pm}^2}{2k_B^2 T^2} - \frac{J_{\pm z}^2}{k_B^2 T^2} + \frac{3J_{zz}^2}{8k_B^2 T^2}\right), \\
\chi_\perp &= \frac{\mu_0\mu_B^2 g_\perp^2}{4k_B T}\left(1 - \frac{3J_\pm}{2k_B T} + \frac{5J_\pm^2}{4k_B^2 T^2} - \frac{J_{\pm\pm}^2}{k_B^2 T^2} - \frac{3J_{\pm z}^2}{4k_B^2 T^2} - \frac{J_{zz}^2}{16k_B^2 T^2}\right).
\end{aligned} \tag{9}$$

In the SU(2)-symmetric point, $J_{zz} = 2J_\pm$, $J_{\pm\pm} = J_{\pm z} = 0$, the two expressions coincide. For the rare-earth local moments with non-Kramers doublets, $g_\perp = 0$ so $\chi_\perp = 0$. In Fig. 2, we plot the magnetic susceptibilities and show the deviation from the simple Curie-Weiss law due to the high order anisotropic terms.

In addition to the simple thermodynamics such as $C_v$ and $\chi$, the magnetic torque measurement is proved to be quite useful in revealing the magnetic anisotropy. Intrinsically, this is because the induced magnetization is generically not parallel to the magnetic field. Thus, when the sample has an anisotropic magnetization, the system would experience a torque $\tau = M \times H = -\partial F/\partial\theta$ in an external magnetic field. The magnetotropic coefficient $k = \partial^2 F/\partial\theta^2$, defined as the second derivative of the free energy to the angle $\theta$ between the sample and the applied magnetic field, can be introduced to quantify such anisotropy. It can be

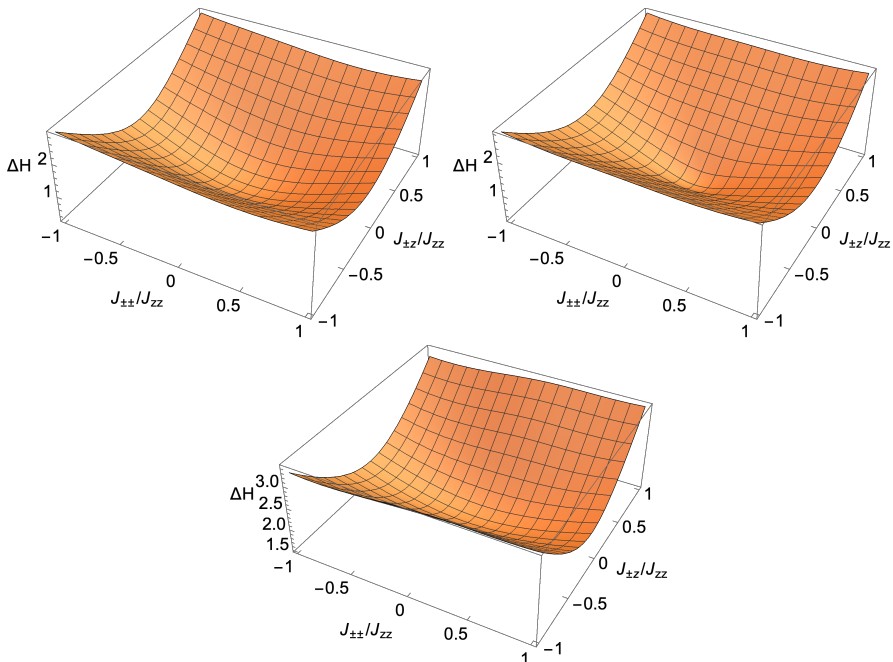

Figure 3: (Color online.) The dependence of $\Delta H$ (defined in equation (14)) on $J_{\pm\pm}/J_{zz}$ and $J_{\pm z}/J_{zz}$, with fixed values of $J_{\pm}/J_{zz}$. Left: $J_{\pm}/J_{zz} = 1$, middle: $J_{\pm}/J_{zz} = 0.2$, right: $J_{\pm}/J_{zz} = -0.5$. In the first two panels, the linewidths are minimal at the "most isotropic" points $J_{\pm\pm} = J_{\pm z} = 0$. In the right panel, we depict the case $J_{\pm}/J_{zz} < 0$, where the in-plane exchange is ferromagnetic. The linewidths $\Delta H$ in the three plots are in the units $\mu_B g(\theta)/\sqrt{2\pi}$.

directly measured using the resonant torsion magnetometry [18]. Under the high temperature expansion, we find the magnetotropic coefficient $k$ is given as

$$
\begin{aligned}
k = \frac{\mu_0^2 \mu_B^2 h^2}{k_B T} \cos 2\theta \Big\{ & \frac{1}{4}(g_\perp^2 - g_\parallel^2) + \frac{3}{16 k_B T}(g_\parallel^2 J_{zz} - 2g_\perp^2 J_\pm) \\
& + \frac{1}{192 k_B^2 T^2}\Big[ -3g_\perp^2(-20J_\pm^2 + 16J_{\pm\pm}^2 + 12J_{\pm z}^2 + J_{zz}^2) \\
& + 6g_\parallel^2(4J_\pm^2 + 4J_{\pm\pm}^2 + 8J_{\pm z}^2 - 3J_{zz}^2) - 2\mu_0^2\mu_B^2 h^2(g_\perp^4 - g_\parallel^4)\Big]\Big\} - \frac{\mu_0^4\mu_B^4 h^4}{96 k_B^3 T^3}\cos 4\theta (g_\perp^2 - g_\parallel^2)^2,
\end{aligned}
$$

(10)

where we have defined $h^2 = h_x^2 + h_y^2 + h_z^2$. Note that there are two different sources of a nontrivial torque magnetometry: the anisotropy of the g-tensor and the anisotropy of the exchange. In the limit of $g_\perp = g_\parallel$, there is still a nonzero contribution to $k$, detailed in Appendix B. The coefficient will only vanish if the Heisenberg limit is further taken: $J_{zz} = 2J_\pm$, $J_{\pm\pm} = J_{\pm z} = 0$.

## 4 Electron spin resonance

In the thermodynamic properties, the leading contributions come from the $J_{zz}$ and $J_\pm$ terms, while the $J_{\pm\pm}$ and $J_{\pm z}$ terms are subleading. Arising from spin-orbital entanglement and completely breaking the U(1) rotational symmetry, these terms play important roles in the potential quantum spin liquid behavior. To resolve them, we now turn to the electron spin resonance.

Electron spin resonance measures the absorption of electromagnetic radiation by a sample subjected to an external static magnetic field. For a SU(2) invariant system, the absorption is completely sharp, *i.e.* described by a delta function located exactly at the Zeeman energy [30]. Therefore, the broadening of the resonance spectrum has to arise from the magnetic anisotropy. To understand the contribution of the anisotropy of the nearest-neighbor spin interaction to the ESR linewidth, we decompose the Hamiltonian Eq. (1) into the isotropic Heisenberg part and the anisotropic exchange part

$$H = J \sum_{\langle i,j \rangle} \mathbf{S}_i \cdot \mathbf{S}_j + H', \tag{11}$$

where the Heisenberg coupling $J = (J_{zz} + 4J_{\pm})/3$, and the anisotropic part

$$H' = \sum_{\langle i,j \rangle} S_i^{\mu} \Gamma_{ij,\mu\nu} S_j^{\nu}. \tag{12}$$

Here $\Gamma_{ij}$ is a traceless and symmetric exchange coupling matrix, satisfying

$$
\begin{aligned}
&\Gamma_{ij,xx} = 2J_{\pm}/3 + (\gamma_{ij} + \gamma_{ij}^*)J_{\pm\pm} - J_{zz}/3, \quad &&\Gamma_{ij,yy} = 2J_{\pm}/3 - (\gamma_{ij} + \gamma_{ij}^*)J_{\pm\pm} - J_{zz}/3, \\
&\Gamma_{ij,zz} = 2J_{zz}/3 - 4J_{\pm}/3, &&\Gamma_{ij,xy} = i(\gamma_{ij} - \gamma_{ij}^*)J_{\pm\pm}, \\
&\Gamma_{ij,yz} = i(\gamma_{ij}^* - \gamma_{ij})J_{\pm z}, &&\Gamma_{ij,zx} = (\gamma_{ij} + \gamma_{ij}^*)J_{\pm z}.
\end{aligned}
\tag{13}
$$

Under the Zeeman term of Eq. (8), the ESR can be computed using the Kubo-Tomita formalism [45], yielding a Lorentzian-shaped spectrum. The corresponding linewidth at high temperatures is given by the second and the fourth moments of the ESR line-shape function [46–48]

$$\Delta H(\theta) = \frac{\sqrt{2\pi}}{\mu_B g(\theta)} \left( \frac{M_2^3}{M_4} \right)^{1/2}, \tag{14}$$

where $\theta$ is again the angle between the external field and the sample, and

$$
\begin{aligned}
g(\theta) &= \sqrt{g_{\parallel}^2 \sin^2\theta + g_{\perp}^2 \cos^2\theta}, \tag{15} \\
M_2 &= \frac{\langle [H', M^+][M^-, H'] \rangle}{\langle M^+ M^- \rangle}, \qquad M_4 = \frac{\langle [H, [H', M^+]][H, [H', M^-]] \rangle}{\langle M^+ M^- \rangle}. \tag{16}
\end{aligned}
$$

$M_2$ and $M_4$ are the second and the fourth moments, respectively, and $M^{\pm} \equiv \sum_i S_i^{\pm}$. The expectation "$\langle \cdots \rangle$" in the above equations is taken with respect to high temperatures. Specifically, we find that

$$
\begin{aligned}
M_2 =& \frac{3}{4}(J_{zz}^2 + 4J_{\pm}^2 + 4J_{\pm\pm}^2 + 10J_{\pm z}^2 - 4J_{\pm}J_{zz}), \\
M_4 =& \frac{3}{4}J_{zz}^4 - \frac{9}{2}J_{zz}^3 J_{\pm} + \frac{57}{8}J_{zz}^2 J_{\pm z}^2 + 15J_{zz}^2 J_{\pm}^2 + 6J_{zz}^2 J_{\pm\pm}^2 - \frac{3}{4}J_{zz}J_{\pm\pm}J_{\pm z}^2 - \frac{93}{4}J_{zz}J_{\pm}J_{\pm z}^2 \\
& - 24J_{zz}J_{\pm}J_{\pm\pm}^2 - 30J_{zz}J_{\pm}^3 + \frac{123}{2}J_{\pm z}^4 + \frac{153}{2}J_{\pm\pm}^2 J_{\pm z}^2 + \frac{39}{2}J_{\pm}J_{\pm\pm}J_{\pm z}^2 + 33J_{\pm}^2 J_{\pm z}^2 \\
& + 15J_{\pm\pm}^4 + 30J_{\pm}^2 J_{\pm\pm}^2 + 24J_{\pm}^4.
\end{aligned}
\tag{17}
$$

The high-temperature ESR linewidths (14), expressed in terms of the different exchanges (17), can be compared with future experiments on the rare-earth based honeycomb magnets in order to extract the anisotropic exchanges. The $J_{zz}$ and $J_{\pm}$ exchanges are easier to indicate from analyzing the experimental data of susceptibilities, as they have lower-order effects. Using their extracted values, one can infer the corresponding $J_{\pm\pm}$ and $J_{\pm z}$ from the ESR linewidth $\Delta H$. In Fig. 3, we depict the three-dimensional plots that explicitly demonstrate the dependence of the ESR linewidth on the anisotropic couplings $J_{z\pm}$ and $J_{\pm\pm}$ for three different choices of $J_{\pm}$.

# 5 Polarized phases

## 5.1 Strong field normal to the honeycomb plane

To further explore the effect of the anisotropic exchange interaction, we study the spin wave excitation with respect to the polarized states under the strong magnetic fields. This is clearly feasible in the current laboratory setting for the rare-earth magnets as the energy scales for them are usually rather small. For the $4d/5d$ magnets, there can be difficulty to achieve as the energy scale over there is much higher, typically by two orders. Our results here are relevant to the inelastic neutron scattering and thermal Hall transport measurements.

We first consider the case of a strong magnetic field in the direction normal to the honeycomb plane such that the system is in the fully polarized paramagnetic phase and all the spins are aligned along the $z$ direction. In this case, the magnon bands carry nontrivial Chern numbers for generic range of parameters, as found in reference [39] in the $J−K−\Gamma−\Gamma'$ presentation. We expand about this fully polarized state using the conventional Holstein-Primakoff transformations of the spin variables [49], which are $S_i^z = S - a_i^\dagger a_i, S_i^+ = a_i, S_i^- = a_i^\dagger$ for sublattice A, and substitute $a \to b$ for sublattice B. $a$ and $b$'s are bosonic operators, $[a_i, a_j^\dagger] = [b_i, b_j^\dagger] = \delta_{ij}$. Keeping only the bilinear terms of bosonic operators and taking the Fourier transformation, we arrive at

$$H = \frac{9N}{4}J_{zz} - 2N\mu_0\mu_B g_\parallel h_\parallel + \frac{1}{2}\Upsilon_{\mathbf{k}}^\dagger \mathcal{H}_{\mathbf{k}}\Upsilon_{\mathbf{k}}, \tag{18}$$

with $\Upsilon \equiv (a_{\mathbf{k}}, b_{\mathbf{k}}, a_{-\mathbf{k}}^\dagger, b_{-\mathbf{k}}^\dagger)^T$, and $N$ being the number of sites in one sublattice. Here we have denoted $k_1 = -\frac{1}{2}k_x + \frac{\sqrt{3}}{2}k_y, k_2 = -\frac{1}{2}k_x - \frac{\sqrt{3}}{2}k_y$, and $k_3 = k_x$ that correspond to the $y$-, $z$- and $x$-bonds, respectively. We further define $f(\mathbf{k}) = \sum_i e^{ik_i}, g_1(\mathbf{k}) = \sum_i e^{-ik_i}\gamma_i, g_2(\mathbf{k}) = \sum_i e^{ik_i}\gamma_i$ and $u \equiv (g_\parallel\mu_0\mu_B h_\parallel - 3J_{zz}/2)$, we then have for $\mathcal{H}_{\mathbf{k}}$ a block form

$$\mathcal{H}_{\mathbf{k}} = \begin{bmatrix} A(\mathbf{k}) & B(\mathbf{k}) \\ B^\dagger(\mathbf{k}) & A^T(-\mathbf{k}) \end{bmatrix}, \tag{19}$$

where we have

$$A(\mathbf{k}) = \begin{bmatrix} u & J_\pm f^* \\ J_\pm f & u \end{bmatrix}, \qquad B^\dagger(\mathbf{k}) = \begin{bmatrix} 0 & J_{\pm\pm}g_1 \\ J_{\pm\pm}g_2 & 0 \end{bmatrix}. \tag{20}$$

All the $J_{\pm z}$ terms are not present. The spin wave dispersion relation for $\mathcal{H}_{\mathbf{k}}$ follows as

$$\begin{aligned}\epsilon(\mathbf{k})^2 = &u^2 + |f|^2 J_\pm^2 - \frac{|g_1|^2 + |g_2|^2}{2}J_{\pm\pm}^2 \\ &\pm [4|f|^2 u^2 J_\pm^2 + \frac{(|g_1|^2 - |g_2|^2)^2}{4}J_{\pm\pm}^4 + (f^*g_1^* - fg_2^*)(f^*g_2 - fg_1)J_\pm^2 J_{\pm\pm}^2]^{1/2},\end{aligned} \tag{21}$$

where only the positive square root of $\epsilon(\mathbf{k})^2$ is taken. Several simple limits of this expression can be checked: (1) in the Heisenberg limit $J_{zz} = 2J_\pm \equiv 2J$, then $\epsilon(\mathbf{k}) = (\frac{g_\parallel\mu_0\mu_B}{2S}h_\parallel - 3J \pm |f|J)^{1/2}$; (2) when only $J_{zz}$ is finite, it reduces to the Ising case $\epsilon(\mathbf{k}) = \frac{g_\parallel\mu_0\mu_B}{2S}h_\parallel - \frac{3}{2}J_{zz}$; if only $J_\pm$ is present, we have a graphene-like dispersion $\epsilon(\mathbf{k}) = \frac{g_\parallel\mu_0\mu_B}{2S}h_\parallel \pm |f|J_\pm$.

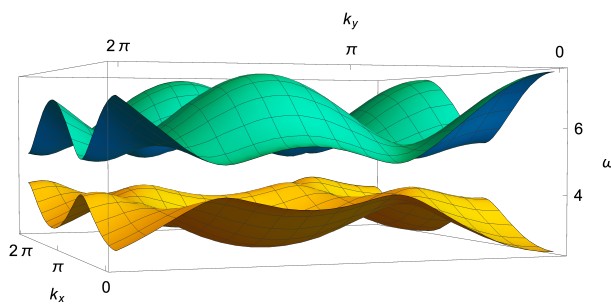

Figure 4: (Color online.) The two spin wave bands $\omega_\pm$ when a strong field in the $z$-direction is applied. The parameters are chosen as $J_{zz} = 1; J_\pm = 0.9; J_{\pm\pm} = 1; J_{\pm z} = 0.3; u = 5$.

At high fields, the results can be simplified by the Schrieffer-Wolff transformation,

$$\tilde{\mathcal{H}}_{\mathbf{k}} = e^W \mathcal{H}_{\mathbf{k}} e^{-W} = \mathcal{H}_{\mathbf{k}} + [W, \mathcal{H}_{\mathbf{k}}] + \frac{1}{2}\big[W, [W, \mathcal{H}_{\mathbf{k}}]\big] + \cdots, \tag{22}$$

with the commutator understood as

$$[X, Y] \equiv X\eta Y - Y\eta X, \tag{23}$$

and $\eta$ is a diagonal matrix with entries $(1, 1, -1, -1)$. Following the treatment of Ref. [39], we choose the transformation to be

$$W = \frac{1}{2u}\begin{pmatrix} 0 & B(\mathbf{k}) \\ -B^\dagger(\mathbf{k}) & 0 \end{pmatrix}, \tag{24}$$

so that up to $O(h_\parallel^{-2})$, we have the $\tilde{\mathcal{H}}_{\mathbf{k}}$ to become $A(\mathbf{k}) \to \tilde{A}(\mathbf{k})$, $B(\mathbf{k}) \to \tilde{B}(\mathbf{k})$,

$$\tilde{A}(\mathbf{k}) = \begin{pmatrix} u - \frac{J_{\pm\pm}^2}{2u}|g_2|^2 & f^* J_\pm \\ f J_\pm & u - \frac{J_{\pm\pm}^2}{2u}|g_1|^2 \end{pmatrix}, \qquad \tilde{B}(\mathbf{k}) = -\frac{J_\pm J_{\pm\pm}}{2u}\begin{pmatrix} f^* g_1^* + g_2^* f & 0 \\ 0 & f^* g_1^* + g_2^* f \end{pmatrix}. \tag{25}$$

At high fields, we can thus ignore $\tilde{B}(\mathbf{k})$ and focus on the $\tilde{A}(\mathbf{k})$ term. Writing $\tilde{A}(\mathbf{k}) = d_0(\mathbf{k})\mathbb{1} + \frac{1}{2}\mathbf{d}(\mathbf{k}) \cdot \boldsymbol{\sigma}$, with the three components being

$$d_1(\mathbf{k}) = 2J_\pm \, Re(f), \qquad\qquad d_2(\mathbf{k}) = 2J_\pm \, Im(f),$$

$$d_3(\mathbf{k}) = \frac{J_{\pm\pm}^2}{2u}(|g_1|^2 - |g_2|^2), \qquad d_0(\mathbf{k}) = u - \frac{J_{\pm\pm}^2}{4u}(|g_1|^2 + |g_2|^2). \tag{26}$$

At each momentum $\mathbf{k}$ we have the eigenvalues

$$\omega_\pm(\mathbf{k}) = d_0(\mathbf{k}) \pm \frac{1}{2}|\mathbf{d}(\mathbf{k})|. \tag{27}$$

The above spin wave bands Eq. (27) do not touch unless $J_\pm = J_{\pm\pm} = 0$, as we have depicted in Fig. 4. We further compute the Berry curvature as follows

$$F_\pm^{xy}(\mathbf{k}) = \pm\frac{i}{2}\left[\frac{\mathbf{d}(\mathbf{k})}{|\mathbf{d}(\mathbf{k})|^3} \cdot \left(\frac{\partial \mathbf{d}(\mathbf{k})}{\partial k_y} \times \frac{\partial \mathbf{d}(\mathbf{k})}{\partial k_x}\right)\right]. \tag{28}$$

This is negative semi-definite in the Brillouin zone. The Chern numbers follow as

$$C_\pm = \frac{1}{2\pi i}\int_{BZ} dk_x dk_y F_\pm^{xy} = \mp 1. \tag{29}$$

This implies the presence of chiral magnon edge states and thermal Hall effect, resulting from the presence of magnon number non-conserving terms $B(\mathbf{k})$ in the Hamiltonian [34, 39]. The edge state for the open boundary condition is depicted in Fig. 5.

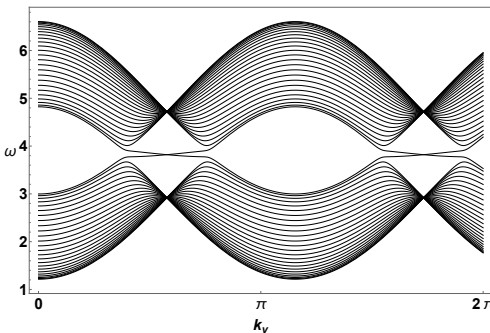

Figure 5: Edge state in a cylindrical geometry. The $y$-direction is periodic, while $x$-direction contains 50 sites. The parameters are $J_{zz} = 1$, $J_\pm = 0.9$, $J_{\pm\pm} = 0.6$, $u = 4$.

## 5.2 Strong field along the honeycomb plane

We now turn to a strong in-plane field, in the $x$-direction. This is relevant for the rare-earth local moments with the usual Kramers doublet, and does not apply to the non-Kramers doublet. The Holstein-Primakoff transformation for sublattice A is modified as $S_i^x = \frac{1}{2} - a_i^\dagger a_i, S_i^y = \frac{1}{2}(a_i + a_i^\dagger), S_i^z = \frac{1}{2i}(a_i - a_i^\dagger)$, and that for sublattice B is obtained by substituting $a$ by $b$. These are again bosonic operators satisfying $[a_i, a_j^\dagger] = [b_i, b_j^\dagger] = \delta_{ij}$. Keeping only the bilinear terms of bosonic operators and taking the Fourier transformation, we obtain

$$H = \frac{9N}{2}J_\pm - 2N\mu_0\mu_B g_\perp h_x + \frac{1}{2}\Upsilon_\mathbf{k}^\dagger \mathcal{H}_\mathbf{k}\Upsilon_\mathbf{k}, \tag{30}$$

$$\Upsilon \equiv (a_\mathbf{k}, b_\mathbf{k}, a_{-\mathbf{k}}^\dagger, b_{-\mathbf{k}}^\dagger)^T. \tag{31}$$

Define $g_3(\mathbf{k}) = e^{ik_1} + e^{ik_2} - 2e^{ik_3}$, $g_4(\mathbf{k}) = e^{ik_1} - e^{ik_2}$ and recall $f(\mathbf{k}) = \sum_i e^{ik_i}$. The $\mathcal{H}_k$ is of the familiar form

$$\mathcal{H}_\mathbf{k} = \begin{bmatrix} A(\mathbf{k}) & B(\mathbf{k}) \\ B^\dagger(\mathbf{k}) & A^T(-\mathbf{k}) \end{bmatrix},$$

but now with the $A, B$ matrices given by

$$A(\mathbf{k})_{11} = A(\mathbf{k})_{22} = v = \mu_0\mu_B g_\perp h_x - 3J_\pm,$$
$$A(\mathbf{k})_{21} = A(-\mathbf{k})_{12} = (\frac{1}{4}J_{zz} + \frac{1}{2}J_\pm)f + \frac{g_3}{4}J_{\pm\pm},$$
$$B(\mathbf{k})_{11} = B(\mathbf{k})_{22} = 0,$$
$$B(\mathbf{k})_{21} = B(-\mathbf{k})_{12} = (-\frac{1}{4}J_{zz} + \frac{1}{2}J_\pm)f + \frac{1}{4}J_{\pm\pm}g_3 + \frac{i\sqrt{3}}{2}J_{\pm z}g_4. \tag{32}$$

Appealing again to the Schrieffer-Wolff transformation with

$$W = \frac{1}{2v}\begin{pmatrix} 0 & B(\mathbf{k}) \\ -B^\dagger(\mathbf{k}) & 0 \end{pmatrix}, \tag{33}$$

then up to $O(h_\perp^{-2})$, we have the effective $\tilde{\mathcal{H}}_k$ to be $A(\mathbf{k}) \to \tilde{A}(\mathbf{k})$, $B(\mathbf{k}) \to \tilde{B}(\mathbf{k})$.

$$\tilde{A}(\mathbf{k}) = \begin{pmatrix} v - \frac{1}{2v}|B(\mathbf{k})_{12}|^2 & A(\mathbf{k})_{12} \\ A(\mathbf{k})_{21} & v - \frac{1}{2v}|B(\mathbf{k})_{21}|^2 \end{pmatrix}, \qquad \tilde{B}(\mathbf{k}) = -\frac{\mathbb{1}}{2v}\big[A(\mathbf{k})_{21}B(\mathbf{k})_{12} + A(\mathbf{k})_{12}B(\mathbf{k})_{21}\big]. \tag{34}$$

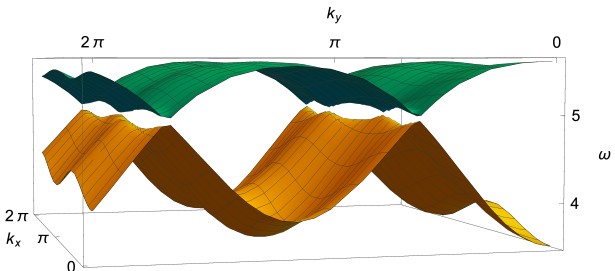

Figure 6: (Color online.) The two spin wave bands $\omega_\pm$ when a strong in-plane field is present. The parameters are chosen $J_{zz} = 1$; $J_\pm = 0.9$; $J_{\pm\pm} = 1$; $J_{\pm z} = 0.3$; $v = 5$.

At high fields $h_\perp$, we can ignore $\tilde{B}(\mathbf{k})$ and focus on the $\tilde{A}(\mathbf{k})$ term. Rewrite $\tilde{A}(\mathbf{k}) = d_0(k)\mathbb{1} + \frac{1}{2}\mathbf{d}(\mathbf{k})\cdot\sigma$, with each component being

$$d_1 = (\frac{1}{4}J_{zz} + \frac{1}{2}J_\pm)Re(f) + \frac{1}{4}J_{\pm\pm}Re(g_3),$$

$$d_2 = (\frac{1}{4}J_{zz} + \frac{1}{2}J_\pm)Im(f) + \frac{1}{4}J_{\pm\pm}Im(g_3),$$

$$d_3 = -\frac{1}{2v}\Big[i\frac{\sqrt{3}}{2}g_4^* J_{\pm z}[(\frac{1}{4}J_{zz} + \frac{1}{2}J_\pm)f + \frac{1}{4}J_{\pm\pm}g_3] + c.c\Big],$$

$$d_0 = v - \frac{1}{2v}[(\frac{1}{4}J_{zz} + \frac{1}{2}J_\pm)^2|f|^2 + \frac{3}{4}J_{\pm z}^2|g_4|^2]. \tag{35}$$

We then arrive at the dispersions $\omega_\pm(\mathbf{k}) = d_0(\mathbf{k}) \pm \frac{1}{2}|\mathbf{d}(\mathbf{k})|$. The spectrum is plotted in Fig. 6. We find both bands have zero Chern numbers, and we have checked for many other parameter choices and also obtained trivial zero Chern number. Thus, the in-plane field magnon band structure is quite distinct from the topological magnon band structure for the normal-plane field case.

# 6 Discussion

We have studied the experimental consequences of the spin-orbital entanglement and the anisotropic spin exchange interactions in the honeycomb rare-earth magnets. These results can be directly compared with the experiments, thereby providing a useful guidance for the future study on candidate systems. One future direction would be to involve higher-lying crystal field states based on the information of specific materials.

One potential rare-earth candidate for the anisotropic honeycomb lattice model is $YbCl_3$ [50], which has a similar crystal structure to that of $RuCl_3$. The $Yb^{3+}$ ions have nearly filled $4f$-orbitals, which, combined with the large crystal fields lead to Kramers doublet ground state manifold. This is modeled as an effective spin-1/2 local moment. Furthermore, its edge-shared octahedral structure gives simple exchange physics that is relatively well-understood according to a microscopic calculation in Ref. [15]. There is very limited information about this material in the literature apart from a very recent work [51]. Comparing their susceptibility data with our calculations, we are able to pin down the exchange parameters $J_{zz} \sim 8K, J_\pm \sim 6K$. Another relevant material is $TlYbS_2$ with AB-stacking triangular structure, which is equivalent to the honeycomb lattice. Recent experiment [52] shows that it has no long-range magnetic order down to $0.4K$, suggesting it to be a frustrated magnet and possible candidate for spin liquid.

In this paper, we have focused the analysis on the honeycomb lattice rare-earth magnets and its anisotropic interaction. It is noticed that, the generic model for the rare-earth honey-

comb magnets contains a Kitaev interaction as one independent exchange interaction out of four. It is thus reasonable for us to consider the possibility of Kitaev materials among the honeycomb rare-earth magnets. In fact most rare-earth magnets have not been discussed along the line of Kitaev interactions, except the first few works [13–15]. In the previous work [13], we have illustrated this observation with the FCC rare-earth magnets. Since many non-honeycomb lattice iridates are claimed as Kitaev materials, it is thus reasonable to consider the rare-earth magnets with other crystal structures to be potential Kitaev materials beyond the previously proposed ones and the honeycomb one here [53, 54]. The reason that these rare-earth magnets contain a Kitaev interaction is due to two facts. The first fact is the spin-orbital-entangled effective spin-1/2 local moment. The second fact is the three-fold rotation symmetry at the lattice site. This symmetry permutes the effective spin components and generates a Kitaev interaction. These two ingredients can be used as the recipe to search for other rare-earth Kitaev materials beyond the honeycomb one.

To summarize, we have focused on rare-earth honeycomb materials with nearest-neighbor interactions and computed the high-temperature thermodynamic properties, ESR linewidth, and spin-wave behaviors as the experimental consequences of the anisotropic spin interaction.

## Acknowledgments

We thank Leon Balents, Ni Ni, and Jiaqiang Yan for conversations. Gang Chen thanks Prof Peng Xue for her hospitality during the visit to Beijing Computational Science Research Center where this work is completed. This work is supported by the Ministry of Science and Technology of China with the Grant No.2016YFA0301001,2016YFA0300500,2018YFE0103200, by from the Research Grants Council of Hong Kong with General Research Fund Grant No.17303819, by the Heising-Simons Foundation, and the National Science Foundation under Grant No. NSF PHY-1748958.

## A Generic spin models and candidate states for higher spins

This spin model in Eq. (1) is designed for effective spin-1/2 local moments. It can be well extended to the high-spin local moments. For the honeycomb lattice with spin-1 local moments, the pairwise spin interaction is given as

$$
\begin{aligned}
H = \sum_{\langle ij \rangle} & J_{zz} S_i^z S_j^z + J_\pm (S_i^+ S_j^- + S_i^- S_j^+) + J_{\pm\pm}(\gamma_{ij} S_i^+ S_j^+ + \gamma_{ij}^* S_i^- S_j^-) \\
& + J_{\pm z}[(\gamma_{ij}^* S_i^+ S_j^z + \gamma_{ij} S_i^- S_j^z) + \langle i \leftrightarrow j \rangle] + \sum_i D(S_i^z)^2.
\end{aligned}
\tag{36}
$$

Because of the larger Hilbert space, a single-ion anisotropy (the $D$-term) is allowed and new states such as the quantum paramagnet can be favored here. Thus, the phase transitions between quantum paramagnet and other ordered phases can be interesting. Further neighbor exchange interaction, if included, could bring more frustration channel than the spin-orbit entanglement induced frustration. It is known that, simple $J_1$-$J_2$ (first neighbor and second neighbor Heisenberg) model on honeycomb lattice could induce spiral spin liquids in two dimensions where the spiral degeneracy has a line degeneracy in the momentum space rather than the surface degeneracy. The presence of the anisotropic interaction in Eq. (36) would overcome the quantum/classical order by disorder effect and lift the degeneracy. In addition to the spin-1 local moments, the model in Eq. (36) also applies to the spin-3/2 systems. Since the honeycomb lattice contains three nearest neighbor bonds, one may consider the possibility

of the AKLT states on the honeycomb lattice where the nearest neighbor bonds are covered with spin singles of the spin-1/2 states and three onsite spin-1/2 spins are combined back to a spin-3/2 local moments.

Here, we have only listed the pairwise spin interactions. Due to the spin-orbital entanglement and the spin-lattice coupling, the effective interaction for the spin-1 and spin-3/2 magnets can contain significant multipolar interactions. A simple example would be the biquadratic exchange $-(\boldsymbol{S}_i \cdot \boldsymbol{S}_j)^2$ that is induced effectively by the spin-lattice coupling. The presence of these multipolar interactions can significantly enhance quantum fluctuation by allowing the system to tunnel more effectively within the local spin Hilbert space and thus create more quantum states such as multipolar ordered phases and quantum spin liquids [55,56].

The relevant physical systems for the spin-1 and spin-3/2 moments would contain the $4d^2, 5d^2, 4d^4, 5d^4$ and $4d^1, 5d^1, 4d^3, 5d^3$ magnetic ions, respectively. The relevant ions can arise from Ru, Mo and even V atoms, where spin-orbit coupling in the partially filled $t_{2g}$ shell is active [55,56].

# B   Details of high temperature expansion

The high temperature expansion requires to take into account the commutation relations between different spin operators on a same site. To this end, we define a vertex function $v_i(n_x, n_y, n_z)$ at each site $i$ following Ref. [43], where $n_x, n_y$ and $n_z$ have to be even integers for the function to be nonzero. Its explicit form can be calculated by introducing the generating function

$$\psi(\xi, \eta, \zeta) = \mathrm{Tr}\,\left[\exp(\xi S^x + \eta S^y + \zeta S^z)\right]. \tag{37}$$

Expanding the exponential and using the definition of $v$, we have

$$\psi(\xi, \eta, \zeta) = 2 \sum_{n_x=0}^{\infty} \sum_{n_y=0}^{\infty} \sum_{n_z=0}^{\infty} \frac{v(n_x, n_y, n_z)}{2^{n_x+n_y+n_z}} \frac{\xi^{n_x} \eta^{n_y} \zeta^{n_z}}{n_x! n_y! n_z!}. \tag{38}$$

On the other hand, by diagonalizing the matrix of the exponential, we have

$$\psi(\xi, \eta, \zeta) = 2 \cosh\left(\sqrt{\xi^2 + \eta^2 + \zeta^2}/2\right). \tag{39}$$

Expanding this and comparing with the previous equation,

$$v(n_x, n_y, n_z) = \frac{[(n_x + n_y + n_z)/2]!}{(n_x/2)!(n_y/2)!(n_z/2)!} \frac{n_x! n_y! n_z!}{(n_x + n_y + n_z)!}. \tag{40}$$

We note this function is symmetric under the permutation of $n_x, n_y, n_z$. The heat capacity is related to the zero-field partition function in the following way

$$C = \frac{1}{N} \frac{\partial E}{\partial T} = \frac{\beta^2}{N} \left[ \frac{1}{Z_0} \frac{\partial^2 Z_0}{\partial \beta^2} - \frac{1}{Z_0^2} \left( \frac{\partial Z_0}{\partial \beta} \right)^2 \right], \tag{41}$$

where we have divided by the number of sites $N$ to get the intensive quantity, and $\beta = 1/k_B T$. $Z_0$ is given by

$$Z_0 = 2^N \left[ 1 + \frac{1}{4}\beta^2 \sum_{\langle ij \rangle} (\frac{1}{8} J_{zz}^2 + J_{\pm}^2 + J_{\pm\pm}^2 + J_{\pm z}^2) \right] + O(\beta^3), \tag{42}$$

where the $2^N$ factor results from the summation over all possible configurations. Susceptibility in direction $a$ can be reduced to the following expectation values of two spin operators,

$$\chi_a = \frac{1}{\beta N}\frac{\partial^2}{\partial h_a^2}\ln Z\big|_{h_a=0} = \frac{\mu_0\mu_B^2 g_a^2}{N Z_0}\beta\langle\sum_{m,n}S_m^a S_n^a\rangle_0. \tag{43}$$

Using the vertex function $v(n_x, n_y, n_z)$ defined above, we obtain for the parallel case,

$$\langle\sum_{m,n}S_m^z S_n^z\rangle_0 = \sum_{\{\mathbf{S}_i\}}\sum_{m,n}S_m^z S_n^z e^{-\beta H}$$

$$=\sum_{\{\mathbf{S}_i\}}\left[\sum_{m,n}S_m^z S_n^z - \beta\sum_{\langle ij\rangle}\sum_{m,n}H_{ij}S_m^z S_n^z + \frac{1}{2}\beta^2\sum_{\langle ij\rangle}\sum_{\langle kl\rangle}\sum_{m,n}H_{ij}H_{kl}S_m^z S_n^z + \text{permutations} + O(\beta^3)\right]$$

$$=\sum_{\{\mathbf{S}_i\}}\left\{\frac{N}{4} - \frac{3N}{16}\beta J_{zz} + \frac{3N}{128}\beta^2 J_{zz}^2 + (\frac{3N^2}{32} - \frac{3N}{16})\beta^2(\frac{1}{8}J_{zz}^2 + J_\pm^2 + J_{\pm\pm}^2 + J_{\pm z}^2)\right.$$

$$+ \frac{\beta^2}{32}(J_\pm^2 + J_{\pm\pm}^2)\sum_{\langle ij\rangle}\left[v_i(2,0,2)v_j(2,0,0) + v_i(2,0,2)v_j(0,2,0) + v_i(0,2,2)v_j(2,0,0)\right.$$

$$\left.+ v_i(0,2,2)v_j(0,2,0)\right]$$

$$+ \frac{\beta^2}{32}J_{\pm z}^2\sum_{\langle ij\rangle}\left[v_i(2,0,2)v_j(0,0,2) + v_i(0,2,2)v_j(0,0,2) + v_i(0,0,4)v_j(2,0,0)\right.$$

$$\left.+ v_i(0,0,4)v_j(0,2,0)\right] + \frac{1}{16}\beta^2\sum_{\langle ij\rangle}(J_{zz}^2 - 2J_{\pm z}^2) + O(\beta^3)\right\}$$

$$=2^N\cdot\frac{N}{4}\left[1 - \frac{3}{4}\beta J_{zz} + \beta^2(\frac{3}{8}J_{zz}^2 - \frac{1}{2}J_\pm^2 - \frac{1}{2}J_{\pm\pm}^2 - J_{\pm z}^2) + \frac{3N}{8}\beta^2(\frac{1}{8}J_{zz}^2 + J_\pm^2 + J_{\pm\pm}^2 + J_{\pm z}^2) + O(\beta^3)\right]. \tag{44}$$

Here, the summation for $\{\mathbf{S}_i\}$ is over the possible configurations of spins on all sites. The notation $H_{ij}$ means the terms in the Hamiltonian for the bond labeled by sites $i, j$; namely, $H = \sum_{\langle ij\rangle}H_{ij}$. "Permutations" on the second line are those with respect to the relative orderings of $H_{ij}$, $H_{kl}$, $S_m^z$ and $S_n^z$. Similarly, for the perpendicular susceptibility, we have

$$\langle\sum_{m,n}S_m^x S_n^x\rangle_0 = \sum_{\{\mathbf{S}_i\}}\sum_{m,n}S_m^x S_n^x e^{-\beta H}$$

$$=\sum_{\{\mathbf{S}_i\}}\left[\sum_{m,n}S_m^x S_n^x - \beta\sum_{\langle ij\rangle}\sum_{m,n}H_{ij}S_m^x S_n^x + \frac{1}{2}\beta^2\sum_{\langle ij\rangle}\sum_{\langle kl\rangle}\sum_{m,n}H_{ij}H_{kl}S_m^x S_n^x + \text{permutations} + O(\beta^3)\right]$$

$$=\sum_{\{\mathbf{S}_i\}}\left\{\frac{N}{4} - \frac{3N}{8}\beta J_\pm + \frac{N-2}{16}\beta^2\sum_{\langle ij\rangle}(\frac{1}{8}J_{zz}^2 + J_\pm^2 + J_{\pm\pm}^2 + J_{\pm z}^2) + \frac{1}{64}\beta^2 J_{zz}^2\sum_{\langle ij\rangle}v_i(2,0,2)v_j(0,0,2)\right.$$

$$+ \frac{1}{32}\beta^2(J_\pm^2 + J_{\pm\pm}^2)\sum_{\langle ij\rangle}\left[v_i(4,0,0)v_j(2,0,0) + v_i(4,0,0)v_j(0,2,0) + v_i(2,2,0)v_j(2,0,0)\right.$$

$$\left.+ v_i(2,2,0)v_j(0,2,0)\right]$$

$$+ \frac{1}{32}\beta^2 J_{\pm z}^2\sum_{\langle ij\rangle}\left[v_i(4,0,0)v_j(0,0,2) + v_i(2,2,0)v_j(0,0,2) + v_i(2,0,2)v_j(2,0,0)\right.$$

$$\left.+ v_i(2,0,2)v_j(0,2,0)\right]$$

$$+\beta^2 \sum_{\langle ij \rangle} \sum_{\langle jk \rangle} \left[ \frac{1}{8} J_\pm^2 + \frac{1}{16} J_{\pm\pm}^2 (\gamma_{ij}\gamma_{jk}^* + \gamma_{ij}^*\gamma_{jk}) + \frac{1}{32} J_{\pm z}^2 (\gamma_{ij}\gamma_{jk} + \gamma_{ij}\gamma_{jk}^* + c.c) \right] + O(\beta^3) \Big\}$$

$$= 2^N \cdot \frac{N}{4} \Big[ 1 - \frac{3}{2}\beta J_\pm + \frac{3N}{8}\beta^2 (\frac{1}{8}J_{zz}^2 + J_\pm^2 + J_{\pm\pm}^2 + J_{\pm z}^2) - \frac{1}{16}\beta^2 J_{zz}^2 + \frac{5}{4}\beta^2 J_\pm^2 - \beta^2 J_{\pm\pm}^2 - \frac{3}{4}\beta^2 J_{\pm z}^2$$

$$+ O(\beta^3) \Big].$$

(45)

One can similarly compute $\langle \sum_{m,n} S_m^x S_n^x \rangle_0$ and arrive at the same expression as above. The cross terms $\langle \sum_{m,n} S_m^x S_n^y \rangle_0$ turn out to be zero.

The magnetotropic coefficient $k$ can be computed using its relationship with the partition function with non-zero external field,

$$k = \frac{1}{N} \frac{\partial^2 F}{\partial \theta^2} = \frac{1}{\beta N} \left[ \frac{1}{Z^2} \left( \frac{\partial Z}{\partial \theta} \right)^2 - \frac{1}{Z} \frac{\partial^2 Z}{\partial \theta^2} \right]. \tag{46}$$

The first term always give higher order terms compared with the second term, while the latter reads,

$$\frac{\partial^2 Z}{\partial \theta^2}$$

$$= -\beta \mu_0 \mu_B \sum_i \langle g_\perp \cos\theta (\cos\varphi S_i^x + \sin\varphi S_i^y) + g_\parallel \sin\theta S_i^z \rangle + \beta^2 \mu_0^2 \mu_B^2 \sum_{i,j} \langle g_\perp^2 \sin^2\theta \cos^2\varphi S_i^x S_j^x$$

$$+ g_\perp^2 \sin^2\theta \sin^2\varphi S_i^y S_j^y + g_\parallel^2 \cos^2\theta S_i^z S_j^z - g_\perp g_\parallel \sin\theta \cos\theta \cos\varphi (S_i^z S_j^x + S_i^x S_j^z)$$

$$- g_\perp g_\parallel \sin\theta \cos\theta \sin\varphi (S_i^z S_j^y + S_i^y S_j^z) + g_\perp^2 \sin^2\theta \sin\varphi \cos\varphi (S_i^x S_j^y + S_i^y S_j^x) \rangle + O(\beta^3)$$

$$= \mu_0^2 \mu_B^2 \beta^2 \cos 2\theta (g_\parallel^2 - g_\perp^2) + \frac{3\mu_0^2 \mu_B^2}{4} \beta^3 \cos 2\theta (2g_\perp^2 J_\pm - g_\parallel^2 J_{zz}) - \frac{\mu_0^4 \mu_B^4}{48} \beta^4 \cos 4\theta (g_\perp^2 - g_\parallel^2)^2$$

$$\times (3N - 2)$$

$$- \frac{\mu_0^2 \mu_B^2}{192} \beta^4 \cos 2\theta \Big\{ 3N(g_\perp^2 - g_\parallel^2) \Big[ 4\mu_0^2 \mu_B^2 (g_\perp^2 + g_\parallel^2) + 3(J_{zz}^2 + 8J_\pm^2 + 8J_{\pm\pm}^2 + 8J_{\pm z}^2) \Big]$$

$$- 4 \Big[ 2\mu_0^2 \mu_B^2 (g_\perp^4 - g_\parallel^4) - 6g_\parallel^2 (-3J_{zz}^2 + 4J_\pm^2 + 4J_{\pm\pm}^2 + 8J_{\pm z}^2) + 3g_\perp^2 (J_{zz}^2 - 20J_\pm^2 + 16J_{\pm\pm}^2$$

$$+ 12J_{\pm z}^2) \Big] \Big\}. \tag{47}$$

The expression above reduces to, in the limit $g_\perp = g_\parallel$,

$$\frac{\partial^2 Z}{\partial \theta^2} \Big|_{g_\perp = g_\parallel} = 2^N \cdot \frac{N}{4} \beta^3 \mu_0^2 \mu_B^2 \cos 2\theta \Big[ -12J_{zz} + 24J_\pm + \beta(7J_{zz}^2 - 28J_\pm^2 + 8J_{\pm\pm}^2 - 4J_{\pm z}^2) \Big]. \tag{48}$$

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
