# Peer review of "Honeycomb rare-earth magnets with anisotropic exchange interactions"

_SciPost Physics, doi:SciPost Phys. Core 3, 004 (2020)_

## Round 1 · Referee Report · Anonymous (Referee 1) · 2020-4-17

Strengths

1- Important theoretical insight for Kitaev material candidates. 2- Connection to experimentally measurable quantities.

Weaknesses

1- Few inconsistencies in the writing and in the notation used.

Report

Luo and Chen theoretically studied rare-earth magnets with anisotropic interactions within the honeycomb lattice. They calculated thermodynamical quantities such as specific heat, susceptibilities, and magnetotropic coefficients, as well as electron spin resonances, under the influence of external magnetic fields. They find the appearance of non-trivial topological magnons in the system, and also conclude that this model could be treated as a Kitaev material due to the appearance of independent Kitaev interactions in the Hamiltonian. Finally, they discuss the possibility that YbCl3 as candidate material and they argue that their calculations should be noticeable in thermal Hall transport experiments.
The study is very interesting and scientifically sound; hence they should be interesting to many scientists working in Kitaev physics and quantum spin liquid proposals. Therefore, the motivations for the study presented in this manuscript are justified.
In my opinion, this paper is suitable for publication in Scipost, after the authors clarify the following minor issues:
1) In the last paragraph of page 1, the authors argue that a small magnetic field is enough for getting a fully polarized state, but then they consider only strong magnetic fields in their calculations. Would the author care elaborate on what is “small” and what is “strong”, by providing experimental orders of magnitudes in order to make these points clear?
2) In Section II, after the parametrization for the Hamiltonian is done (Eq. 2), please briefly explain what the terms $J$, $K$, and $\Gamma$ are. Especially considering that one of the main findings is that the rare-earth magnets can be treated as Kitaev models, but the Kitaev term is not introduced to the reader.
3) For the reader’s benefit, please after Eqs. 5 and 7 define what $k_B$, $\mu_0$, and $\mu_B$ are. Also, around Eq. 7 define that $h$’s are the components of the external magnetic field.
4) In the caption of Figure 2, write what the dashed-blue and solid-black lines are.
5) At the very end of Section IV, when discussing ESR shown in Fig. 3 please provide a longer discussion on what is shown in Fig. 3, and highlight the difference among the panels rather than saying that is ESR is anisotropic-coupling dependent.
6) First paragraph of Section V.A, the line “…energy scales for them are usually rather small.” should have a reference at the end. Please provide it.
7) Title of Section V.B should read “Strong field along the honeycomb plane”, since “in” could be misinterpreted.
8) Typos: - Page 3, left column, last paragraph: “there is” should read as “this is”. – Page 3, right column, last line: “a traceless” should read as “is a traceless”. – Page 7, first paragraph: “spin-obital” should read as “spin-orbital”.
9) After Eq. A1, please define what $D(S_i^z)$ is.
10) After Eq. B5, please define what this $\beta$ is, since in Eq. 2 $\beta$ was something else.
11) Why the perpendicular susceptibility of Eq. B9 only contains $S^x$ terms and not $S^y$? Please elaborate.
  • validity: top
  • significance: high
  • originality: top
  • clarity: high
  • formatting: perfect
  • grammar: excellent

Author:  Zhu-Xi Luo  on 2020-05-14  [id 825]

(in reply to Report 1 on 2020-04-17)

1) Q: In the last paragraph of page 1, the authors argue that a small magnetic field is enough for getting a fully polarized state, but then they consider only strong magnetic fields in their calculations. Would the author care elaborate on what is “small” and what is “strong”, by providing experimental orders of magnitudes in order to make these points clear?

A: "A small magnetic field" means the absolute magnitude of the field is small, while the ``strong magnetic fields'' in the calculation means that compared with the exchange energy scale of the system, the field is strong. See also the reply to question 6. The exchange energy scale for rare-earth magnets is usually much smaller than $d$ electrons, thus a conventionally small magnetic field would be sufficient to generate a large effect.

2) Q: In Section II, after the parametrization for the Hamiltonian is done (Eq. 2), please briefly explain what the terms J, K, and $\Gamma$ are. Especially considering that one of the main findings is that the rare-earth magnets can be treated as Kitaev models, but the Kitaev term is not introduced to the reader.

A: Added explanations below equations (1) and (3).

3) Q: For the reader’s benefit, please after Eqs. 5 and 7 define what $k_B$, $\mu_0$, and $\mu_B$ are. Also, around Eq. 7 define that h’s are the components of the external magnetic field.

A: Done as suggested.

4) Q: In the caption of Figure 2, write what the dashed-blue and solid-black lines are.

A: Done as suggested.

5) Q: At the very end of Section IV, when discussing ESR shown in Fig. 3 please provide a longer discussion on what is shown in Fig. 3, and highlight the difference among the panels rather than saying that is ESR is anisotropic-coupling dependent.

A: We have expanded the discussions in the caption of figure 3 as well as below eqn. (17).

6) Q: First paragraph of Section V.A, the line “…energy scales for them are usually rather small.” should have a reference at the end. Please provide it.

A: The exchange interactions in rare-earth materials are typically two orders of magnitude smaller than in transition metal magnets with similar inter-atomic distances. For example, those of the 5d material Na2IrO3 [Phys. Rev. Lett. 113, 107201 (2014)] and 4d material RuCl_3 [Scientific Reports volume 6, 37925 (2016)] are O(1) meV, while those of YbMgGaO4 [Phys. Rev. Lett. 115, 167203, (2015)] and Ba3Yb2Zn5O11 [Phys.Rev.B, 98:054408, (2018)] are O(10^{-2}) meV. We have added the general comment to the beginning of section V. There is no satisfactory comprehensive literature that compares the 4d/5d versus rare-earth honeycomb materials (the latter are not widely studied at all), so we prefer not to cite the separate references, which may be distractive. However, we are able to combine our calculations and experimental data of YbCl3 [arXiv: 1903.03615] and find its exchange to be O(10^{-2}) meV. We have added this to the discussion section.

7) Q: Title of Section V.B should read “Strong field along the honeycomb plane”, since “in” could be misinterpreted.

A: Revised as suggested. Thanks.

8) Typos: - Page 3, left column, last paragraph: “there is” should read as “this is”. – Page 3, right column, last line: “a traceless” should read as “is a traceless”. – Page 7, first paragraph: “spin-obital” should read as “spin-orbital”.

A: Revised as suggested. Thanks.

9) Q: After Eq. A1, please define what D(Szi) is.

A: $D(S_i^z)^2$ is the single-ion anisotropy as explained in the first sentence after equation (A1). We edited the text to make it more explicit.

10) Q: After Eq. B5, please define what this $\beta$ is, since in Eq. 2 $\beta$ was something else.

A: Done. Thanks for catching that.

11) Q: Why the perpendicular susceptibility of Eq. B9 only contains Sx terms and not Sy? Please elaborate.

A: Revised as suggested. The $\langle Sy Sy\rangle$ terms give the same result as the $\langle Sx Sx\rangle$ terms, and the cross-terms don't contribute.

---

## Round 1 · Referee Report · Edson Vernek (Referee 2) · 2020-4-18

Strengths

1) The topic/material is very attracting;

2) The manuscript is overall well written;

3) The results sound physically correct.

Weaknesses

1) The manuscript lacks focus.

2) The richness of the results is rather poorly discussed.

Report

I have read with interest the manuscript entitled "Honeycomb rare-earth magnets with anisotropic exchange interactions" by Zhu-Xi Luo and Gang Chen. The authors investigate the anisotropic effect induced by spin-orbit coupling in a system composed of rare earth magnets on a honeycomb lattice. By employing the known Kitaev-Heisenberg model, the authors explore the high temperature regime of the magnetic susceptibility, specific heat, electron spin resonance and the spin-wave band structure in the strong polarized regime. Despite being fairly well written, I think the manuscript lacks focus. It is hard to grasp the "central message" of the paper. For example, the authors state in the discussion, Sec. VI: "In this paper, we have focused the analysis on the honeycomb lattice rare-earth magnets and its anisotropic interaction". Since other studies have addressed this problem (for instance, their Ref. 39), this sentence suggests that the authors fail providing an appropriate focus in the paper.

In view of the above, I cannot recommend publication of the manuscript in the present form. As far as the results concern, those presented in section V are the one I appreciate most. Therefore, I think the authors should be able to make improvements on their manuscript to overcome these weakness.

Below I list some question and comments that should be clarified in a future version of the manuscript:

1) What is precisely the physical quantity shown in Fig. 3? Is it the one defined in Eq. (13)? In the text it is hard to make a connection.

2) In connection to the question above, I do not quite understand the sentence "the ESR linewidth for a Lorentzian-shaped spectrum is ... "

3) In the set of Eqs. (17), $\Gamma_{zz}$ is not given. Should I infer $\Gamma_{zz}=-\Gamma_{xx} -\Gamma_{yy}$ from the traceless properties of the tensor $\Gamma_{ij}$?

4) I find interesting the result shown in Fig. 5. In connection to it, I have some questions:

a) Being this band structure topologically non-trivial, are the gapless states protected by some symmetry? which one?

b) Would be worthwhile to investigate the symmetry class of the effective Hamiltonian in this regime, if it has not been done in some reference I'm no aware of?

Minor points:

1) In the introduction, the authors state "Due to the possible proximity to Kitaev physics, these materials were referred as Kitaev materials." The word proximity here causes confusion because of the known term "proximity effect" typically found in topological materials. I would suggest perhaps replacing proximity by similarity.

2) At the beginning of Sec. III, the author could indicate Appendix B where the calculation of thermodynamics is shown with some detail.

Requested changes

No specific change, see Report.

  • validity: good
  • significance: good
  • originality: ok
  • clarity: good
  • formatting: good
  • grammar: good

Author:  Zhu-Xi Luo  on 2020-05-14  [id 826]

(in reply to Report 2 by Edson Vernek on 2020-04-18)

1) Q: What is precisely the physical quantity shown in Fig. 3? Is it the one defined in Eq. (13)? In the text it is hard to make a connection.

A: Yes, it is the one in eqn. (13). We have made this point more explicit in the caption of figure 3.

2) Q: In connection to the question above, I do not quite understand the sentence "the ESR linewidth for a Lorentzian-shaped spectrum is ... "

A: We added more explanations in the new version.

3) Q: In the set of Eqs. (17), $\Gamma_{zz}$ is not given. Should I infer $\Gamma_{zz}=-\Gamma_{xx}-\Gamma_{yy}$ from the traceless properties of the tensor $\Gamma_{ij}$?

A: Yes, we have only listed the independent components. But to avoid confusion, we have added the zz component into the equation.

4) Q: I find interesting the result shown in Fig. 5. In connection to it, I have some questions:

a) Being this band structure topologically non-trivial, are the gapless states protected by some symmetry? which one?

b) Would be worthwhile to investigate the symmetry class of the effective Hamiltonian in this regime, if it has not been done in some reference I'm no aware of?

A: It is BdG Hamiltonian without effective time reversal symmetry (the $B(k)$ terms break it), so the system is in symmetry class D, and the gapless states are protected by the effective charge conjugation symmetry $U^\dagger H^* U=-H$.

Minor points:

1) Q: In the introduction, the authors state "Due to the possible proximity to Kitaev physics, these materials were referred as Kitaev materials." The word proximity here causes confusion because of the known term "proximity effect" typically found in topological materials. I would suggest perhaps replacing proximity by similarity.

A: Indeed the word "proximity" can cause confusion. "Similarity" should work, but we tended to mean something more like "relevance". We have changed it in the new version.

2) Q: At the beginning of Sec. III, the author could indicate Appendix B where the calculation of thermodynamics is shown with some detail.

A: Revised as suggested, thanks.

---

## Round 1 · Referee Report · Anonymous (Referee 3) · 2020-4-21

Report

In this paper, the authors study several properties of a general effective spin-1/2 model on the honeycomb lattice. On the basis of a high-temperature expansion, they discuss the susceptibility, torque magnetometry, and the ESR bandwidth. They also discuss the nature of the excitations in the high field, fully polarised state. The idea to turn to rare-earths to realise experimentally the Heisenberg-Kitaev model is interesting, but it is not clear to me what the impact of this paper will be, and also to which extent some of the results are new. So, I would like the authors to address the following points:

  • What is specific to rare earths? The model of Eq. (1) has already been studied in the context of iridates, right? Is it just the fact that this model is expected to be a more accurate approximation than for iridates because one can neglect further neighbour interactions in the case of rare earths? When I saw the title of the paper, I was expecting some derivation (or at least some discussion) of the form of Eq. (1) for rare earths, and to get a feeling for which parameters can be expected to dominate, but I did not find anything of that sort.

  • I think it would be nice to comment on the qualitative implications of Eq. (9). Usually, torque magnetometry relies on the anisotropy of the g-tensor to measure the magnetization. Here, the torque response will be different because of the other sources of anisotropy.

  • ESR linewidth: What is the take-home message? By measuring the line width, one can check the parameters of the model with the help of Eqs. (13) and (17)? This should be specified.

  • Polarized phase with a strong field normal to honeycomb phase: What is new with respect to Ref. 39? Maybe I missed something, but the result looks very similar to that of Ref. 39, with just another parametrisation.

  • validity: high
  • significance: ok
  • originality: ok
  • clarity: high
  • formatting: excellent
  • grammar: good

Author:  Zhu-Xi Luo  on 2020-05-14  [id 827]

(in reply to Report 3 on 2020-04-21)

Q: What is specific to rare earths? The model of Eq. (1) has already been studied in the context of iridates, right? Is it just the fact that this model is expected to be a more accurate approximation than for iridates because one can neglect further neighbour interactions in the case of rare earths? When I saw the title of the paper, I was expecting some derivation (or at least some discussion) of the form of Eq. (1) for rare earths, and to get a feeling for which parameters can be expected to dominate, but I did not find anything of that sort.

A: Yes, one can neglect further neighbor interactions in the case of rare earths. As for the dominating parameters for general rare-earth magnets, due to the low angular momentum of these materials, the interactions between the rare-earth ions are generically quantum, with all the (symmetry-allowed) exchanges being potentially significant.
A detailed derivation of the exchange parameters requires the knowledge of the environment and chemical information for specific materials. Different rare-earth ions would be different from each other. Instead, we take a more phenomenological view and rely more on the experimental input to extract the parameters. For example, using our calculations and the experimental data of susceptibilities in Ref[51], we find that $J_{zz}\sim 8K$ and $J_{\pm}\sim 6K$. We have added this into the discussion section.

Q: I think it would be nice to comment on the qualitative implications of Eq. (9). Usually, torque magnetometry relies on the anisotropy of the g-tensor to measure the magnetization. Here, the torque response will be different because of the other sources of anisotropy.

A: Thanks for pointing that out. We have added some remarks after equation (9).

Q: ESR linewidth: What is the take-home message? By measuring the line width, one can check the parameters of the model with the help of Eqs. (13) and (17)? This should be specified.

A: Yes, thanks. We have expanded the discussions in the caption of figure 3 and below eqn. (17).

Q: Polarized phase with a strong field normal to honeycomb phase: What is new with respect to Ref. 39? Maybe I missed something, but the result looks very similar to that of Ref. 39, with just another parametrisation.

A: This is easy to be achieved in rare-earth magnets and is an indispensable experiment to be carried out for these materials. Although it is only one small section in our paper, We feel it is needed to have a full story for the understanding and the exploration of the experimental consequences from the anisotropic interactions between rare-earth local moments.

---

## Round 6 · Referee Report · Anonymous (Referee 3) · 2020-6-2

Report

The authors have properly taken care of my questions and suggestions. I think that the paper can now be published.

---

## Round 6 · Referee Report · Edson Vernek (Referee 2) · 2020-6-9

Report

The authors have properly addressed all my comments and clarified all my questions. Likewise, great attention has been given by the authors to the reports of other referees, with detailed responses. As a result, the authors have produces a more clear and improved version of their manuscript, which I can recommend publication in this Journal.

---

## Round 6 · Author Response

In this new version, we mainly expand the discussions for the setup of the Hamiltonian, the magnetometry and the electron spin resonance results. Other revisions involve explanations of notations and corrections of typos.

---

## Round 6 · List of Changes

1. Added two sentences below equations (1) and (3) to further explain the setup.
  2. Below Eqn (9), added more comments on the magnetometry results.
  3. Added another expression for $\Gamma_{ij,zz}$ to equation (12).
  4. In the caption of figure 3 and below Eqn (17), added more explanations about the ESR results.
  5. Added a reference [45] above Eqn (13) and rephrased the sentences.
  6. Added a remark at the beginning of section V about the scale of exchange.
  7. Added an estimation of the YbCl3 exchange scales in the second paragraph of the discussion section.
  8. Added a remark about high-temperature expansion below equation (B9).

Smaller edits: 9. The second paragraph in the introduction section, changed "proximity" to "relevance". 10. At the beginning of section III, specified that Appendix B contains more details. 11. Below equations (5) and (7), added more explanations of notations. 12. The last paragraph of the last column on page 3, corrected a typo "there is"->"this is". 13. In the caption of figure 2, added more explanations of notations. 14. Below Eqn (11), corrected a typo "a traceless"->"is a traceless". 15. Changed "in" to "along" in the title of section V. B. 16. Corrected a typo "spin-obital" -> "spin-orbital" in the second paragraph of left column on page 7. 17. Added explanation of notation below equation (A1) and (B5).

---

## Editorial Decision

published